# PROTOTYPE GENERATION: ROBUST FEATURE VISUALISATION FOR DATA INDEPENDENT INTERPRETABILITY

## ABSTRACT

We introduce *Prototype Generation*, a stricter and more robust form of feature visualisation for model-agnostic, data-independent interpretability of image classification models. We demonstrate its ability to generate inputs that result in natural activation paths, countering previous claims that feature visualisation algorithms are untrustworthy due to the unnatural internal activations. We substantiate these claims by quantitatively measuring similarity between the internal activations of our generated prototypes and natural images. We also demonstrate how the interpretation of generated prototypes yields important insights, highlighting spurious correlations and biases learned by models which quantitative methods over test-sets cannot identify.

## 1 INTRODUCTION

Interpretability techniques have become crucial in the era of increasingly powerful artificial intelligence (AI) systems (OpenAI, 2023; Anthropic, 2021; Anil et al., 2023; Touvron et al., 2023). As AI models continue to outperform human benchmarks across numerous tasks and domains, the importance of understanding their decision-making processes has never been more pressing. This is particularly true for black-box deep learning models, which are increasingly employed across various industries ranging from healthcare to autonomous vehicles (Bohr & Memarzadeh, 2020; Elallid et al., 2022). Apart from safety concern in these high-stakes domains, EU law requires certain AI systems to comply with the 'right to explanation' (Goodman & Flaxman, 2017) making interpretability crucial for business operations.

Over the past decade, various methods have been developed to improve human understanding of complex AI models. Techniques such as LIME (Ribeiro et al., 2016), SHAP (Lundberg & Lee, 2017), CAM (Oquab et al., 2015) and Network Dissection (Bau et al., 2017; Zhou et al., 2018) have targeted local interpretability, offering explanations of model decisions for individual data points. However, these techniques cannot provide a global understanding of *what a model has learned overall*, which is necessary for comprehensive analysis and trust in automated systems.

In this work, we focus on feature visualisation (Olah et al., 2017) as a powerful interpretability tool able to extract such holistic insights from arbitrary neural networks. Despite its promise, feature visualisations have not been without criticism. Past research has pointed out the disparity between internal processing of feature visualisations as compared to other natural images (Geirhos et al., 2023) by observing path similarity. We discuss these criticisms further in Section 2.

Addressing these limitations, we introduce *Prototype Generation* in Section 3, a robust visualisation tool that not only contains determining features for any given class but also maintains equal or better path similarity with natural images. Our experiments using Resnet-18(He et al., 2015) and InceptionV1(Szegedy et al., 2014) show that prototypes generated using our method are highly similar to natural images in terms of internal path activations.

Understanding the model at a global level helps in identifying systemic biases, uncovering spurious correlations, and potentially refining the model for better performance and fairness. We use prototype generation to discover undesired correlations and identify failure modes on unseen data in Section 4, demonstrating how our method provides *data-independent* insights, removing the need

for time-consuming manual inspection of training datasets to subjectively identify unwanted biases. Through this, our contribution serves the broader goal of enhancing global data-independent interpretability in deep learning models, thereby making them more transparent, accountable, and trustworthy.

## 2 RELATED WORK

Feature visualisation is a method to extract information from a model about what it has learned (Molnar, 2021; Mordvintsev et al., 2015; Erhan et al., 2009; Olah et al., 2017; Nguyen et al., 2016). Unlike local interpretability methods that focus on individual predictions, feature visualisation is a *global* interpretability method that aims to visualise the features that different neurons inside a model have learned to respond to. Observing feature visualisations to understand model behaviour is a data-independent approach to interpretability, allowing for qualitative assessment of a model's internal logic irrespective of any test dataset – and so, can be used to find failure modes irrespective of whether examples of those failures exist in a test set. This technique works by generating an input $\hat{x}$ that maximises a chosen output logit or internal activation (in this case, output logit $c$ with respect to model $h$): $\hat{x} = \arg\max_x h_c(x)$. Feature visualisation has been used for a number of purposes, such as identifying specialised circuits within neural networks (Olah et al., 2020) and understanding the learned features of individual nodes or groups of nodes (Olah et al., 2018).

Despite its utility, feature visualisation is not without its detractors. One prominent line of criticism comes from Geirhos et al. (2023), arguing that the visualisations may not truly represent what the model has learned, and so cannot be reliably used to predict its behaviour on unseen data in the future. These criticisms are substantiated by experiments that manipulate feature visualisations to produce misleading or contradictory representations without changing the model's decision-making process. They also introduce the path similarity metric to quantify this. This metric measures the similarity between internal activation 'paths' caused by two different inputs across the layers of a neural network. If two inputs excite similar neurons, this leads to a high path similarity between these two inputs. The measure of similarity chosen by Geirhos et al. (2023) is Spearman's rank order correlation (referred to as spearman similarity (SS) in the rest of this paper). This path similarity metric is used to show the disparity between internal activations in response to natural images versus feature visualisations of the same class.

Geirhos et al. (2023) also provide a proof that feature visualisation cannot formally guarantee trustworthy results, claiming that without additional strong assumptions about the neural network feature visualisations can't be trusted. However, this is also the case with any existing evaluation metric – on a given test set, two models may perform exactly alike, but there is always the possibility that they will differ on some unknown future input. Therefore, we argue that feature visualisation based approaches should not be seen as a magic bullet – but rather as an important and practically useful complement to quantitative assessment metrics.

In the sections that follow, we show that feature visualisations of a specific kind – *prototypes* – generated using our method contain key features for the class they represent, and maintain a consistent path similarity with natural images. By doing so, we overcome some of the limitations previously associated with feature visualisation.

## 3 PROTOTYPE GENERATION

For a given model $M$, we define a prototype $P$ as an input that maximally activates the logit corresponding to $c$, while keeping the model's internal activations in response to that input close to the distribution of 'natural' inputs. Let $\mathbb{I}$ represent the set of all possible natural inputs that would be classified by model $M$ as belonging to class $c$. We aim to generate a prototype $P$ such that it aggregates the representative features of a majority of inputs in $\mathbb{I}$. Formally, we posit that the activations $\mathbf{A}_P$ of $P$ are 'closer' to the mean activations $\mathbf{A}_{\mathbb{I}}$ of all $I \in \mathbb{I}$ than any individual natural image $I$ across all layers $\mathbb{L}$ in $M$. We measure 'closeness' between $\mathbf{A}_P$ and $\mathbf{A}_{\mathbb{I}}$ using two metrics: L1 distance and spearman correlation.

We use spearman similarity as per Geirhos et al. (2023) to allow for direct comparison of our methods with their published work. We also use the L1 distance to further substantiate this comparison.

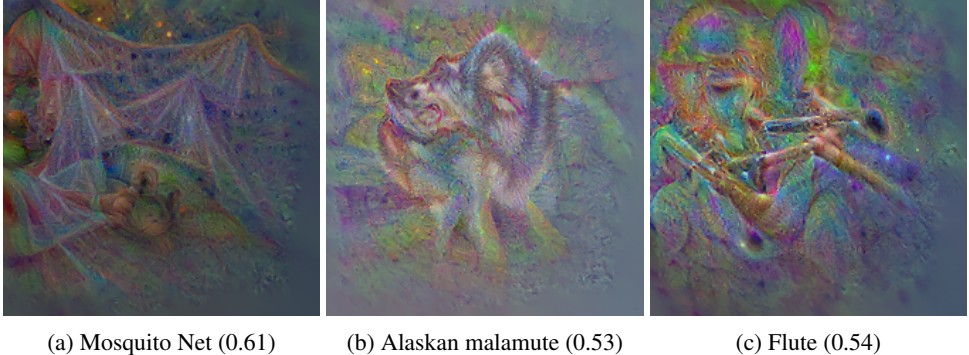

(a) Mosquito Net (0.61)      (b) Alaskan malamute (0.53)      (c) Flute (0.54)

Figure 1: Example prototypes generated by our method for the ImageNet classes Mosquito Net, Alaskan malamute and Flute with their average spearman similarity across all layers of Resnet-18 denoted in brackets

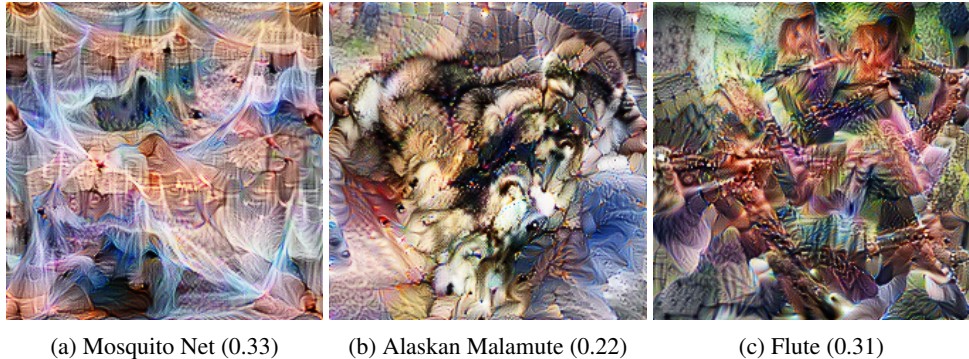

(a) Mosquito Net (0.33)      (b) Alaskan Malamute (0.22)      (c) Flute (0.31)

Figure 2: Example prototypes generated by Olah et al. (2017)'s method for the ImageNet classes Mosquito Net, Alaskan malamute and Flute with their average spearman similarity across all layers of Resnet-18 denoted in brackets

Calculating spearman similarity involves ranking the activations in terms of magnitude, whereas calculating L1 distance preserves the magnitude of activations. Since the input images are subject to preprocessing and so belong to a set training distribution, the magnitude of activations is relevant information that provides a more complete picture of path similarity.

Using L1 distance our formal assertion is that,

$$\sum_{l \in \mathbb{L}} |\mathbf{A}_{\mathbb{I}_l} - \mathbf{A}_{\boldsymbol{P}_{c_l}}| \leq \sum_{l \in \mathbb{L}} |\mathbf{A}_{\mathbb{I}_l} - \mathbf{A}_{\boldsymbol{I}_l}| \tag{1}$$

For the rest of the paper we will denote $\sum_{l \in \mathbb{L}} |\mathbf{A}_{\mathbb{I}_l} - \mathbf{A}_{\boldsymbol{P}_{c_l}}|$ as $D_P$ and $\sum_{l \in \mathbb{L}} |\mathbf{A}_{\mathbb{I}_l} - \mathbf{A}_{\boldsymbol{I}_l}|$ as $D_{\boldsymbol{I}}$.

Denoting spearman similarity as SS, our formal assertion is that:

$$SS(\mathbf{A}_{\mathbb{I}_l}, \mathbf{A}_{\boldsymbol{P}_{c_l}}) \geq SS(\mathbf{A}_{\mathbb{I}_l}, \mathbf{A}_{\boldsymbol{I}_l}), \forall l \in L, \forall \boldsymbol{I} \in \mathbb{I} \tag{2}$$

If both of these conditions are satisfied, we can confidently assert that prototype P shows prototypical qualities of the class $c$, and contains features representative of the model's understanding of that class.

### 3.1 OUR METHOD

Existing feature visualisation methods aim to generate an input that maximally activates a selected neuron inside a neural network. Prototype generation is similarly a technique that generates an

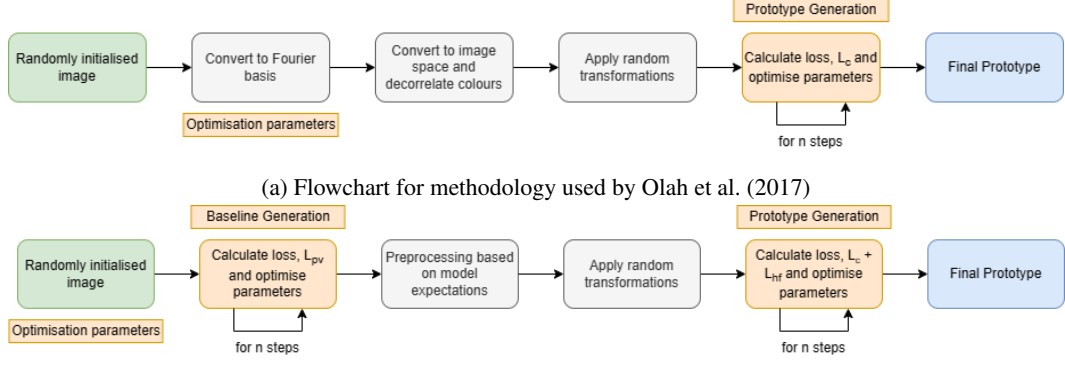

(a) Flowchart for methodology used by Olah et al. (2017)

(b) Flowchart of our method

Figure 3: Comparison between our method and feature visualisation method proposed by Olah et al. (2017)

input, but with the objective of maximally activating a selected output logit (rather than an internal activation), as shown in Figure 1. This positions prototype generation as a specialised form of feature visualisation, distinguished by its focus on class-specific logits rather than internal activations. When a generated input satisfies the criteria we have laid out, remaining within the domain of 'natural' inputs and capturing representative features of its corresponding class, we term it as the model's 'learned prototype' for that class. Here, 'ideal representation' is used to signify that this learned prototype encapsulates what the model perceives as the most representative or 'ideal' features for categorising inputs into that particular class.

Our approach differs from the existing feature visualisation methodology in a number of ways, as shown in Figure 3,

We compare our implementation with the publicly available *Lucent* (Greentfrapp, 2018) library – the PyTorch implementation of the methodology proposed by Olah et al. (2017).

Both implementations begin with a randomly initialised image. Lucent converts this randomly initialised image to the Fourier basis, but we find (as shown later) that this causes the resulting feature visualisations to be unrepresentative of natural class features. In contrast, we do not optimise in the Fourier basis, instead optimising the input pixels directly. We first optimise to minimise what we call probability variance loss, $L_{pv}$ to generate a baseline input. This loss ensures that the output logits for our input image are balanced i.e. the input image has roughly an equal chance of being predicted to be a member of any class. Our preprocessing steps vary depending on the model's expectations; for models trained on ImageNet, this involves a normalisation shift using the mean and standard deviation of the ImageNet training set – of whatever preprocessing the model expects for inference. Additionally we apply random affine transformations to constrain the optimisation process and discourage the generation of out-of-distribution (OOD) adversarial inputs – we further discuss the effect of these transformations in Appendix A. Lucent uses similar random transformations, but does not tune them for path similarity. The difference in the resultant prototypes for *Lucent* and our method can be seen clearly by comparing Figures 1 and 2.

We define two losses: $L_c$, the negative of the logit for the desired class; and $L_{hf}$, the high-frequency loss that penalises large differences between adjacent pixel values. We use both $L_c$ and $L_{hf}$ to define our combined loss whereas Olah et al. (2017) employ only $L_c$.

## 3.2 EXPERIMENTS

We assess the prototypes generated by observing how closely the prototype's activations mirror the average activations of natural images in the same class. We quantify closeness between activations by calculating L1 distance and spearman similarity as defined in Section 2. Appendix A contains information about hyperparameters and other implementation details.

**L1 distance.** We generate prototypes for 11 random classes from Resnet-18 and InceptionV1 and collect 100 random images from each of these classes. We use these prototypes and images to calculate average $D_P$ and average $D_I$ across all 11 classes. In the case of Resnet-18 we find that for the 67 layers in the model, $D_P$ is lower than $D_I$ for 55/67 i.e. 82.1% of Resnet-18 layers. For InceptionV1, we find that $D_P$ is lower than $D_I$ for 150/223 layers i.e. 67.2% of all layers. Figure 4 shows $D_P$ and $D_I$ across all layers in Resnet-18 and InceptionV1. We also plot $D_{I_{dc}}$ where $I_{dc}$ denotes the set of all images that belong to different classes than the class $P$ is generated for. It is clear to see that $D_P$ is lower than both $D_I$ and $D_{I_{dc}}$ for most of the layers for prototypes generated from both Resnet-18 and InceptionV1 showing that the prototypes approximately satisfy our formal assertion related to L1 distances as specifed in Equation 1. For the majority of the model's layers, generated prototypes results in activations that are closer to the mean *natural* activation, than any individual natural input image.

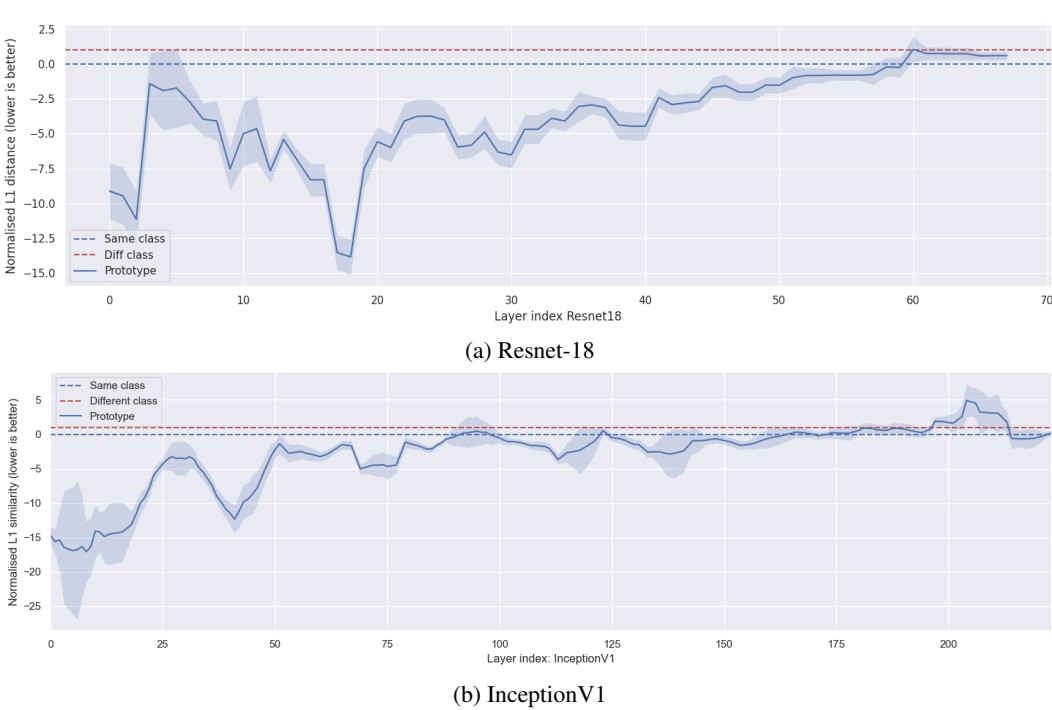

(a) Resnet-18

(b) InceptionV1

Figure 4: **Comparison of L1 Distance**. The L1 distances are normalised such that 0 corresponds to the mean L1 distance between natural image activations of the same class and 1 corresponds to the mean L1 distance between natural image activations of different classes.

**Path similarity per layer.** To characterise the path similarity of our generated prototypes, we select 11 random classes from the ImageNet dataset.

We approximate $A_{\mathbb{I}}$ by averaging the activations of 100 randomly selected images from each of the 11 classes. For each class $c$, we generate a prototype $P$, capture its activations $A_P$, and also capture activations from the individual images in $\mathbb{I}$. To alow for direct comparison of our results with those reported by Geirhos et al. (2023), we also select 100 random images from other classes and capture their activations denoted by $A_{I_{dc}}$. Our raw results consist of three sets of spearman similarity scores between,

- Approximated $A_{\mathbb{I}}$ and $A_P$
- Approximated $A_{\mathbb{I}}$ and $A_I$, averaged across all $I \in \mathbb{I}$
- Approximated $A_{\mathbb{I}}$ and $A_{I_{dc}}$, averaged across all $I_{dc} \in \mathbb{I}_{dc}$

This raw data is normalised such that 1 corresponds to the spearman similarity obtained by comparing natural images of the same class, and 0 corresponds to the spearman similarity obtained by comparing images of one class against images of different classes. To reduce noise in our plots showing

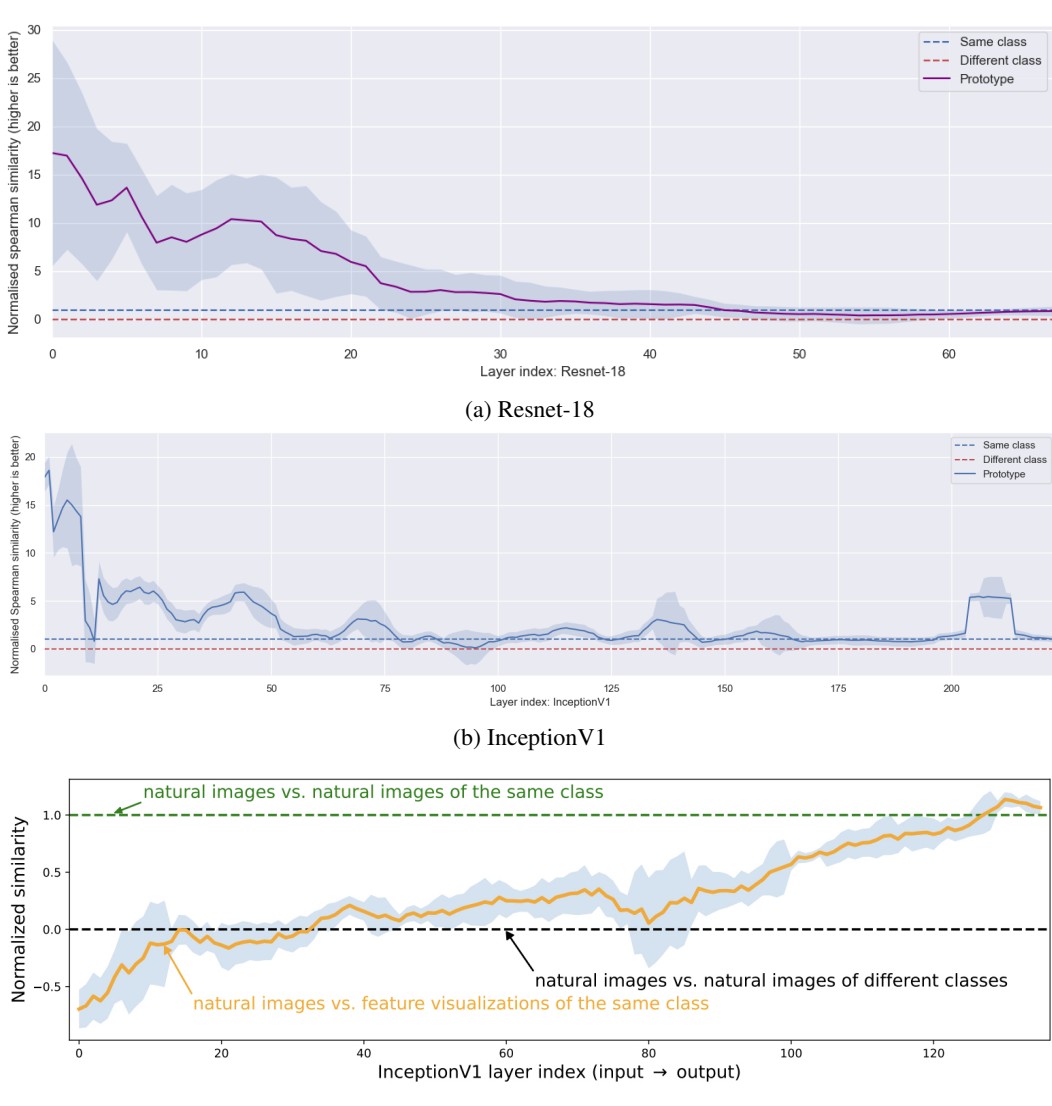

(a) Resnet-18

(b) InceptionV1

(c) InceptionV1, results from Geirhos et al. (2023).

Figure 5: **Comparison of Spearman Similarity**. Spearman similarities are normalised such that 1 corresponds to the spearman similarity between natural image activations of the same class and 0 corresponds to the spearman similarity between natural image activations of different classes. Here we show examples on two different networks, and for comparative purposes also provide the results of the same experiment from Geirhos et al. (2023). Note that our method produces super-normal results, with early layer activations from prototypes being closer to the mean natural activation than any natural input.

spearman similarity between $A_{\mathbb{I}}$ and $A_P$, we smooth the curve using *scipy.ndimage.convolve* with a window size of 10 for the mean values, and 5 for the standard deviations.

Figure 5 shows the normalised path similarity obtained by making the above comparisons. Our experimental results show that these prototypes support our formal assertion for Spearman similarity specified in Equation 2 for 38/67 i.e. 56.7% of all layers and have higher path similarity than natural images for all layers of Resnet-18 averaged across 11 different classes. In the case of InceptionV1 we find that our formal assertion holds for 147/224 i.e 65.6% of all layers in InceptionV1 averaged across the same 11 classes.

Table 1: Comparison of average spearman similarity

|  | Average spearman similarity |
|---|---|
| Prototype $P$ | $0.54 \pm 0.06$ |
| Same class images $\mathbb{I}_c$ | $0.50 \pm 0.05$ |
| Diff class images $\mathbb{I}_{dc}$ | $0.41 \pm 0.06$ |

(a) Resnet-18

|  | Average spearman similarity |
|---|---|
| Prototype $P$ | $0.56 \pm 0.07$ |
| Same class images $\mathbb{I}_c$ | $0.50 \pm 0.05$ |
| Diff class images $\mathbb{I}_{dc}$ | $0.40 \pm 0.06$ |

(b) InceptionV1

Using our raw results, we also quantify the mean spearman similarity across all layers of Resnet-18 and InceptionV1 in Table 1. We can see that the average spearman similarity of our generated prototypes is higher than other natural images belonging to the same class on average, for both Resnet-18 and InceptionV1.

## 4 PROTOTYPE INSIGHTS

Since our method generates prototypes that have high path similarity with natural images, 3 we might expect to be able to better understand *what* models have learned about given classes simply by observing their generated prototypes for those classes. Here follows a case study to test whether information present in our prototypes can reliably predict model behaviour on unseen data. We focus on prototypes for two Imagenet classes: the academic gown prototype and the mortarboard prototype, generated by Resnet-18, as shown in Figure 6. We experimented with a diversity objective as in Olah et al. (2017) to generate varied prototypes but found limited differences in prototypes, we show the results of the diversity objective in Section C.

**Hypothesis 1:** Resnet-18 will have higher accuracy classifying lighter skinned people wearing academic gowns than darker skinned people wearing academic gowns.

This hypothesis emerges from the observation that the academic gown prototype in Figure 6a shows a lighter-skinned face prominently. We test this hypothesis by observing the performance of Resnet-18 on two different sets of images, one containing lighter skinned people wearing academic gowns and the other containing darker skinned people wearing academic gowns. We collect 50 random images for each of these sets from the internet, taking care to maintain a general parity between the two sets in image quality, setting and size. As shown in Table 2, lighter-skinned people wearing academic gowns are more likely to be classified as the academic gown class than darker-skinned people.

**Hypothesis 2:** Resnet-18 is likely to misclassify images of mortarboards as academic gowns, if the images have a mortarboard and a face in them.

By observing differences in the mortarboard and academic gown prototypes, we see that the mortarboard prototype has a much weaker representation of a face, compared to the academic prototype. This leads us to hypothesise that an image containing both a mortarboard and a face is likely to be misclassified as an academic gown. To test this hypothesis we again observe the performance of

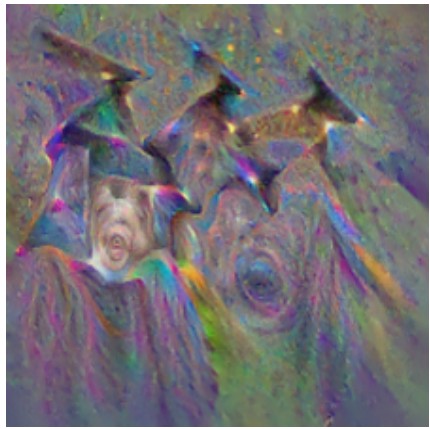

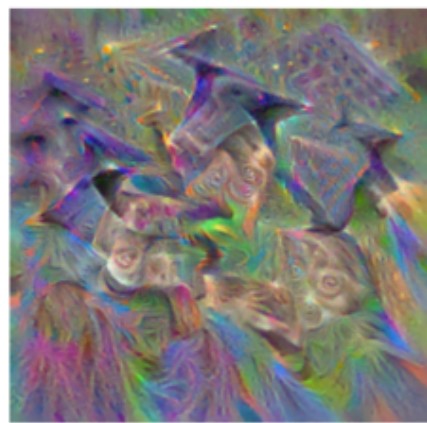

(a) Academic Gown Prototype                    (b) Mortarboard Prototype

Figure 6: Example prototypes from Resnet-18

Table 2: Comparison of Resnet-18 performance on light and dark skinned people wearing academic gowns along with the probability of prediction for academic gowns(AG)

|  | Accuracy | Probability(AG) |
|---|---|---|
| **Lighter-skinned people** | 72.5% | 0.62 |
| **Darker-skinned people** | 60% | 0.54 |

Resnet-18 on a set of images containing mortarboards with no faces and a set of images containing mortarboards and a face. We ensure that the mortarboards with faces have no presence of academic gowns. Results were again as expected, Table 3 shows that mortarboards with faces are more likely to be misclassified as academic gowns.

Table 3: Comparison of Resnet-18 performance on mortarboards with and without faces along with the probability of prediction for mortarboards(MB) and academic gowns(AG)

|  | Accuracy | Probability(MB) | Probability(AG) |
|---|---|---|---|
| **Mortarboard without face** | 92.3% | 0.77 | 0.05 |
| **Mortarboard with face** | 70.5% | 0.67 | 0.25 |

Inspection of these prototypes directly hints at biases embedded in the training data. For instance, the academic gown prototype's lighter-skinned face prominence suggests that the training dataset might have had an over-representation of lighter-skinned individuals – and inspection of ImageNet training dataset shows that this is indeed the case. This imbalance, when unaddressed, could lead to real-world consequences where certain demographic groups consistently experience lower accuracy, perpetuating biases. The differences observed between prototypes can further shine light on potential areas of misclassification. Using the mortarboard example, understanding how the model interprets and prioritises features can identify when and why an image might be misclassified; this model seems to be great at classifying mortarboards in a vacuum, but the inclusion of facial features leads to misclassification. Crucially, all of this is done without reference to any test set of data, meaning that our insights are not constrained only to misclassifications contained within that limited test set. In both these cases, we didn't have to comb through the entire dataset but rather observing the prototypes provided us with a data-independent way of understanding Resnet-18's behaviour on unseen data.

While metrics on a test set can provide a broad overview of a model's performance *with respect to that test set*, they often don't provide the granularity needed to understand *why* a model might be likely to fail on future, unseen data. Prototypes can meaningfully augment existing evaluation metrics in a number of ways:

- **Identifying dataset bias.** If a model shows bias, as in the lighter-skinned academic gown prototype, this points at bias in the data. Armed with this knowledge, the dataset can be modified or augmented to remove this bias, and the model retrained to improve performance on underrepresented classes or features.

- **Spotting spurious correlations.** By comparing prototypes for closely related classes, one can discern which features are given undue importance, enabling deeper understanding of model failures due to the presence of potentially misleading features.

- **Rapid Iteration.** Model developers can generate prototypes during training, spotting issues like biases or potential misclassifications early in the process. These insights also enable targeted data augmentation, necessitating the collection and preprocessing of data samples specific to correcting a problem in the model.

## 5 DISCUSSION

The primary advantage of our methodology is its ability to furnish insights into what a model has learned. In situations where the stakes are high – medical diagnoses, financial predictions, or autonomous vehicles, to name a few – a deeper comprehension of what a model has learned is important for safety.

Our prototypes allow us to essentially engage in an iterative feedback loop that continually enhances a model's performance:

- **Prototype Generation.** Initially, we generate prototypes to visualize the model's understanding of different classes.

- **Insight Extraction.** Once these prototypes are available, they can reveal specific biases or tendencies in the model's learning. As shown in Section 4, if the prototype of an 'academic gown' predominantly features a lighter-skinned individual, it highlights a potential bias in the model's understanding of that class.

- **Targeted Retraining.** Based on the insights derived from the prototypes, targeted retraining can be conducted. Using our earlier example, the model can be retrained with a more diverse set of images representing 'academic gowns', thus rectifying its inherent bias.

Furthermore, if a model is underperforming on a specific class and the reason is not immediately clear from the validation data, generating a prototype can shed light on the shortcomings in its learning. Moreover, interpretability techniques of this kind make *knowledge discovery* possible – that is, if we are able to train a model to perform a task that humans cannot, we can use interpretability to understand what patterns it has identified in its training data that we are unaware of, and thereby gain new insights about that data.

**Future Work** While our method has shown promise, it's essential to acknowledge its limitations. We cannot, as of now, provide a formal proof that feature visualisation of this nature will consistently offer useful insights across all use cases and models. Geirhos et al. (2023) also raise the issue of fooling circuits, and demonstrate that it is possible to make visual changes in the prototype that can still maximally activate a given class logit without containing any representative features of the class, which we do not address.

Although we use path similarity as an evaluation metric to gauge the natural look of prototypes, there are some limitations associated with the metric such as ignoring outliers. This is one of the reasons we don't use the path similarity metric to directly optimise prototypes to have high path similarity with natural images. Rather we tune our parameters and optimisation process to lead to high path similarity. The diversity objective mentioned in Section 4 is underexplored and has the potential to generate prototypes that can capture the entire range of features relevant for a class including outliers allowing for more robust analysis.

We wish to address these limitations in future work, and expand our analysis with further case studies of models deployed in the real world, with reference to both bias detection and knowledge discovery. We also aim to benchmark *Prototype Generation* using tests such as the one introduced by Casper et al. (2023) further showing the robustness of our method.

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

## A  EXPERIMENTAL DETAILS

**Choice of classes.** The 11 randomly chosen classes for our experiment in terms of Imagenet class indices were [12, 34, 249, 429, 558, 640, 669, 694, 705, 760, 786].

**Optimisation parameters.** The randomly initialised image is initialised using torch.rand(3, 224, 224) and the optimisation process uses the Adam optimiser with a constant learning rate of 0.05 over 512 optimisation steps.

**Selecting regularisation parameters.** Generating a feature visualisation that maximises a certain output logit related to class $c$ without any constraints will result in adversarial images that have no representative features of $c$ (Erhan et al., 2009). To guide the optimisation process towards creating feature visualisations that maintain the natural structure of images belonging to $c$ we need to utilise some constraints in the form of regularisation. The concept of regularisation originates from the broader field of optimisation and is crucial for preventing overfitting.

During the prototype generation process we need to be cautious while implementing regularisation. An unprincipled approach can lead to undesirable outcomes, such as creating visualisations that are aesthetically pleasing but lack meaningful information about what the model has truly learned during training. In contrast, foregoing regularisation may result in visualisations that contain high-frequency noise or adversarial inputs, which are not only hard to interpret but also unlikely to yield useful insights. This brings us to the delicate balance we aim to strike: optimising the feature visualisation in such a way that it remains useful and informative while avoiding misleading or un-interpretable outputs. Figure 7 shows the increasing effect of regularisation on generated prototypes, starting from no regularisation at the left and increasing towards the right, for prototypes generated using Resnet-18 for the goldfish class.

We find that path similarity can work as a reliable measure of features that conform to the natural structure found in images belonging to a certain class in the training set. Regularisation can take many forms (Nguyen et al., 2015; Mordvintsev et al., 2015) but our focus here will be on two specific forms of regularisation: high-frequency penalisation and random affine transformations.

High-frequency penalisation aims to suppress unnecessary details and noise, facilitating a cleaner, more interpretable visualisation. Transformation robustness, on the other hand, ensures that minor alterations in the input space do not result in significantly different visualisations, thus maintaining consistency and reliability. We perform high-frequency penalisation to minimise large variation in magnitude of adjacent pixel values across all channels and apply it by adding the loss $L_{hf}$ described earlier in Section 3 to our overall loss. We also ensure transformation robustness using random affine transforms made up of random scale, random translate and random rotation transforms. The parameters of random scale, random translate and random rotation guide the degree of freedom of the random affine transform and lead to varying visual characteristics of our generated prototype.

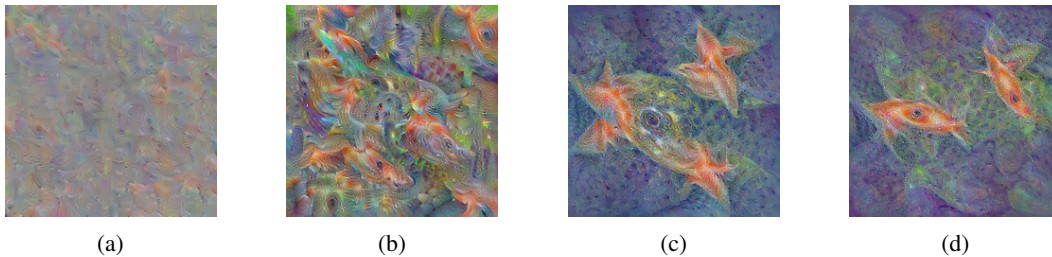

|     (a)     |     (b)     |     (c)     |     (d)     |

Figure 7: Effect of increasing regularisation (from left to right) for the goldfish prototype

To find regularisation parameters that lead to reliable prototypes, we test the effect of random affine transforms on the spearman similarity between prototypes generated using a particular set of regularisations, and average natural image activations from the training set. We test a set of 125 random

affine transforms with varying parameters for random scale, random translate and random rotation transforms,

- Random scale: None, (0.9, 1.1), (0.8, 1.2), (0.7, 1.3), (0.5, 1.5)
- Random rotation: None, 30, 60, 90, 180
- Random translate: None, (0.05, 0.05), (0.1, 0.1), (0.2, 0.2), (0.5, 0.5)

Table 4 shows the top 5 regularisation parameters we discovered, in terms of path similarity calculated using spearman similarity between the $A_P$ (created using these regularisation parameters) and $A_{\mathbb{I}}$, where $\mathbb{I}$ contains 100 natural images belonging to the goldfish class, averaged over all the layers of our model of choice, Resnet-18. The prototype generated using the best regularisation can be seen in Figure 7d.

Table 4: Spearman similarity of top five affine transformation protocols with respect to average natural image activations.

| Scale | Rotation | Translate | Average path similarity |
|:---:|:---:|:---:|:---:|
| (0.7, 1.3) | 180 | (0.5, 0.5) | 0.564 |
| (0.5, 1.5) | 30 | (0.5, 0.5) | 0.563 |
| (0.8, 1.2) | 30 | (0.5, 0.5) | 0.562 |
| (0.5, 1.5) | 60 | (0.1, 0.1) | 0.561 |
| (0.7, 1.3) | 30 | (0.5, 0.5) | 0.560 |

We choose the regularisation parameters that lead to the best average path similarity with the parameters of scale, rotations and translation set at (0.7, 1.3), 180 and (0.5, 0.5) respectively.

## B    RAW EXPERIMENTAL RESULTS

Figures 4 and 5 show the normalised plots for our experimental results comparing L1 distance and Spearman Similarity of our generated prototypes with natural images. In this section, we want to show the raw results before normalisation for our path similarity experiments. Figure 8 shows how close the similarity values are to each other for most part of the network before diverging heavily at the end of the network.

## C    RESULTS OF DIVERSITY OBJECTIVE

We use a pixel-based diversity objective to generate pixel-level differences in the prototypes rather than the cosine difference of activations from a given neuron as in Olah et al. (2017). This leads to imperceptible pixel differences in the prototypes rather then leading to prototypes that show diverse features. Figure 9 shows this change over different classes, although they look exactly the same to human eyes, they lead to different classification probabilities showing that the changes are impreceptible.

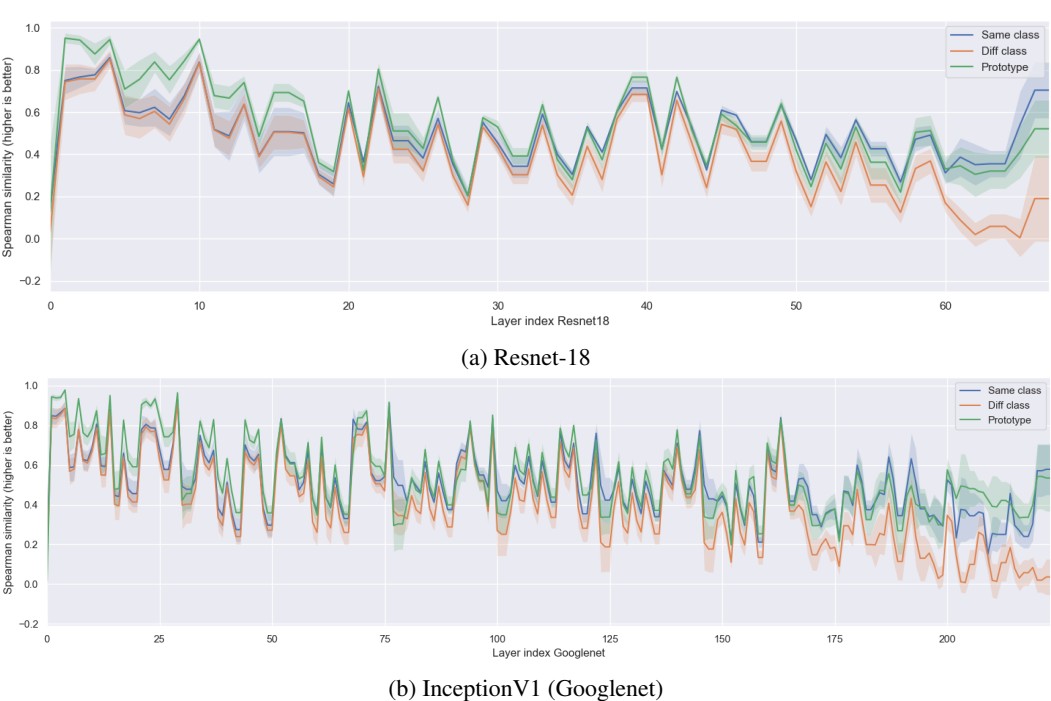

(a) Resnet-18

(b) InceptionV1 (Googlenet)

Figure 8: Raw value plots for spearman similarity comparisons

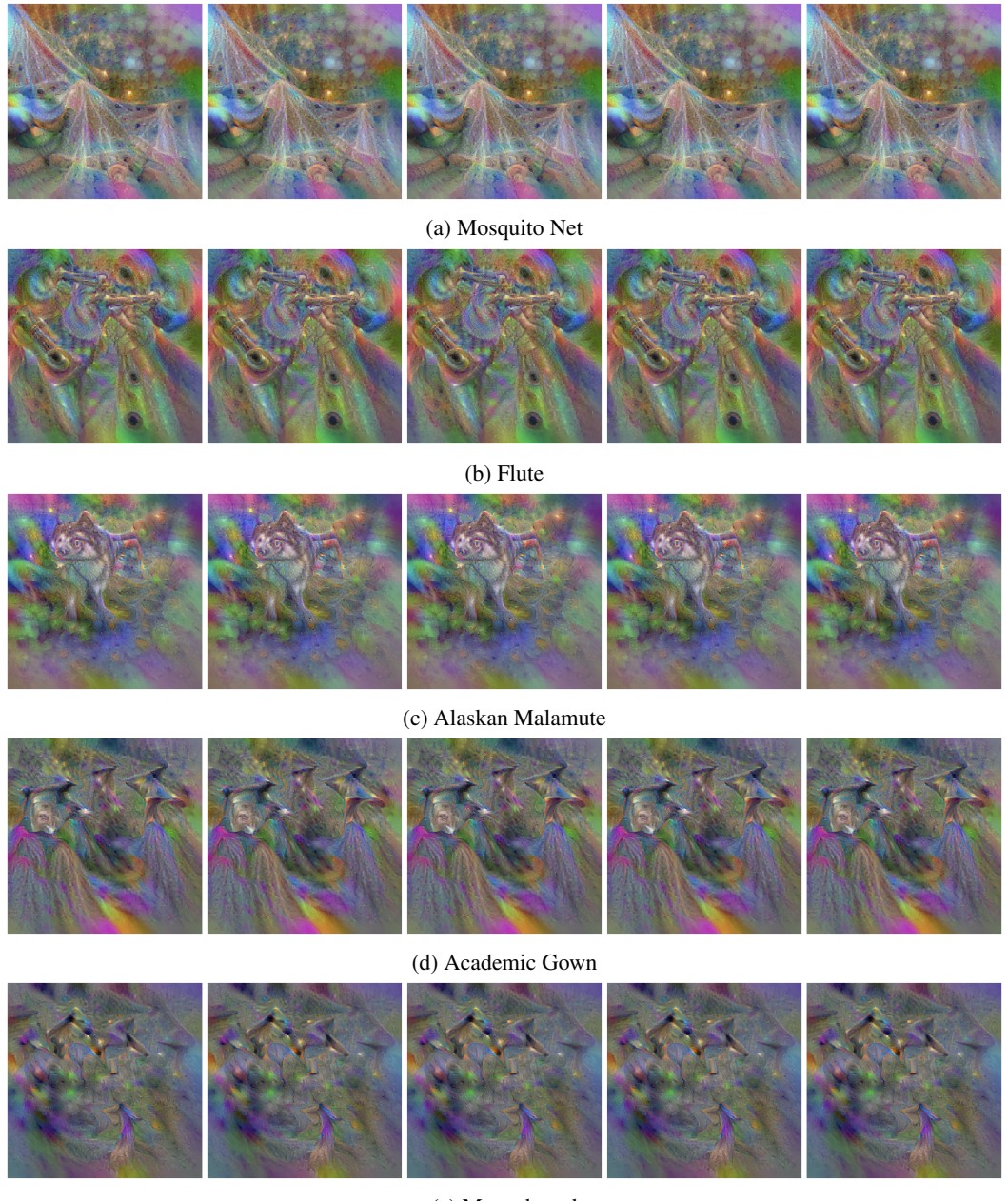

(a) Mosquito Net

(b) Flute

(c) Alaskan Malamute

(d) Academic Gown

(e) Mortarboard

Figure 9: Imperceptible differences using diversity objective

