# OpenReview forum: "Prototype Generation: Robust Feature Visualisation for Data Independent Interpretability"
_ICLR.cc/2024/Conference — Submitted to ICLR 2024_

### Official Review · Reviewer_5oJb · 2023-10-30

**Soundness:** 3 good
**Presentation:** 2 fair
**Contribution:** 3 good
**Rating:** 5
**Confidence:** 4

**Summary:**

The paper proposes a methodology called Prototype Generation for enhancing the interpretability of deep learning models. The authors demonstrate the effectiveness of this approach in identifying biases, spotting spurious correlations, and enabling rapid iteration in model development. The paper also discusses the advantages of this methodology in terms of understanding what a model has learned and facilitating targeted retraining. However, the authors acknowledge the limitations of their method and suggest future work to address these limitations.

**Strengths:**

Overall, the paper presents an interesting approach to improving the interpretability of deep learning models through Prototype Generation. The methodology is well-explained, and the examples provided effectively demonstrate its potential benefits. The paper also highlights the importance of interpretability in high-stakes applications and the need for a deeper comprehension of model behavior.

1. The paper addresses an important problem in the field of deep learning - interpretability.
2. The Prototype Generation methodology is well-described and provides meaningful insights into model behavior.
3. The examples and case studies presented in the paper effectively demonstrate the usefulness of prototypes in identifying biases and understanding model failures.
4. The discussion on targeted retraining and the iterative feedback loop adds practical value to the proposed methodology.

**Weaknesses:**

1. The limitations of the method are acknowledged but not thoroughly discussed. It would be helpful to provide more insights into the potential challenges and drawbacks of Prototype Generation.
2. The paper could benefit from a more detailed comparison with existing interpretability techniques to highlight the novelty and advantages of the proposed approach.

**Questions:**

Please help to check the weakness.

---

> ### Author Response · Authors · 2023-11-21
>
> We thank the reviewer for their thoughtful comments and suggestions.
>
> We have expanded our Future Work section to show more future direction directly targeting many limitations brought up by the reviewers.

---

### Official Review · Reviewer_fFHp · 2023-10-31

**Soundness:** 2 fair
**Presentation:** 1 poor
**Contribution:** 1 poor
**Rating:** 1
**Confidence:** 4

**Summary:**

In this paper, the authors proposed a method called "prototype generation" to visualize the prototypical input (called the "prototype") that would maximally activate the logit of a particular class c. To find such a prototype for each class c for a given neural network, the method starts from a baseline image (generated by optimizing a probability variance loss, which encourages the generated baseline image to have an equal probability to be classified into any class by the network), and then optimizing the baseline image by minimizing the negative logit of the desired class c and the high-frequency loss that penalizes large differences between adjacent pixel values. The expected result is a prototypical input image for class c that maximizes the logit for class c and is smooth and "natural" looking. The authors also performed experiments using a trained ResNet-18 and a trained InceptionV1.

**Strengths:**

- The authors made an attempt to formalize the notion of "path similarity" between a generated prototype and a "natural" image, using L1 distance or spearman similarity.

**Weaknesses:**

- The proposed method lacks novelty. It is exactly the same as activation maximization applied to the logit of each class c, with regularizations to smooth generated images. This idea has been well explored in past work (e.g., Nguyen et al., 2016).
- There is only one prototype generated per class. In reality, each class could have multiple prototypical images.
- While there is an attempt to make generated prototypes more "natural," they do not actually look natural.
- While the authors made an attempt to formalize the notion of "path similarity" between a generated prototype and a "natural" image, this notion is not used to generate prototypes.

**Questions:**

N/A.

**Details Of Ethics Concerns:**

N/A.

---

> ### Author Response · Authors · 2023-11-21
>
> We thank the reviewer for their comments and thoughts.
>
> We overturn previous claims by Geirhos et al. 2023 [1] about the disparity between internal activations of natural activations and feature visualizations by introducing Prototype Generation, a technique that does lead to natural internal activations. We do this by adding in carefully tested objectives and regularisations for path similarity. This is highly non-trivial.
>
> While the notion of path similarity is not used to generate prototypes directly, we used this to test our regularisation hyperparameters as shown in Appendix A.
>
> We have experimented with using path similarity as a metric to generate prototypes with limited success pointing to redundancies in oversubscribing to this objective.
>
> We acknowledge the need for a diversity objective and it’s implications in the new Future Work section.
>
> [1] Robert Geirhos, Roland S. Zimmermann, Blair Bilodeau, Wieland Brendel, and Been Kim. Don’t trust your eyes: on the (un)reliability of feature visualizations, July 2023.

---

### Official Review · Reviewer_WSWR · 2023-11-06

**Soundness:** 3 good
**Presentation:** 3 good
**Contribution:** 2 fair
**Rating:** 5
**Confidence:** 4

**Summary:**

The paper introduces a technique for global interpretability of image classification networks via prototype generation. The main idea is to optimize for an input image (called a prototype for a class) which maximizes the logit corresponding to that class while keeping model parameters constant. The authors then determine a ‘good’ prototype by measuring the spearman correlation between activations of the prototype across all layers ($A_P$) vs the activations of a sample of images from  the class across all layers ($A_I$). They also use L1 distance as another distance metric to measure similarities on $A_P$ and $A_I$.

Experiments are performed on pertained ResNet-18 and Inception V1 on ImageNet. The authors visualize the prototypes on the ‘academic gown’ and ‘mortarboard’ (graduation cap) classes in ImageNet. The gown prototype shows a patch resembling a face with a light skin and the authors conclude that this would result on the model being biased against darker skinned people wearing gowns. This is confirmed via experiments on a hold out test set which shows 12.5% higher performance on lighter skinned individuals.

**Strengths:**

1. Significance of the problem: The paper introduces a technique to very important problem in the interpretability community (easy-to-understand global interpretability).
2. The ideas in the paper are simple to understand and the intuition is well explained.
3. Novelty: Global interpretability via visualization for image classification models has not been done before to my knowledge. I think this makes the technique interesting.

**Weaknesses:**

1. My main concern lies with the qualitative nature of the findings presented in the paper and the resulting ambiguity in the explanations. The authors claim that “the academic gown prototype in Figure 6a shows a lighter-skinned face prominently”. Here 6a refers to the image of the prototype for ‘academic gown’.  Just by looking at that image, I cannot definitively say that the light colored patch is definitely a human face. Without looking at images in the dataset, one cannot assume that there is no other object that shows up in a prototype. Since one of the main claims of the paper is that their method allows us to explain models without combing through the dataset, I’m unconvinced about the practical gain the authors claim their method provides.
2. The second experiment in the paper is about the ‘mortarboard’ (graduation cap) class where the authors look at the prototype and conclude that the model would get confused if they saw faces with caps in the image instead of just caps. Again, the prototype images only contain light colored patches so it is impossible to conclude that they are faces with certainty unless you are explicitly looking for faces in the first place. Thus, in this case, the user already had an explanation in mind and they are only looking at the model for a confirmation of their pre-existing biases rather than a novel explanation.
3. Experiments not comprehensive enough: The authors report experiments on two classes from ImageNet (no justification was given for why these two were chosen specifically). To make sure their method works, we would need to know if this method generalizes to more classes.
4. Details lacking: At several points while reading the paper, I felt there were some significant details lacking. For example: Equations for the loss are not stated (what is the high frequency loss?). Similarly, the authors report having adapted the visualization technique from Olah et al. It would be nice to have a brief description of technique so that the paper is more self contained. I was also confused by Fig 3 which talks about optimizing parameters. I was under the impression that we are optimizing the input image w.r.t output logits. What parameters are we talking about here? Please correct me if I am wrong.
5. Minor: Notation in equation 1 is not clearly explained. I think I understand what each of the terms mean from the context but clearly spelling out what each term means (including the subscripts) is important for good readability.
6. Limitations: Since the activations of the prototype is compared to the average of a sample of images from the input class distribution, the prototype is going to be biased towards covering the average of the most common features and ignore outliers.

**Questions:**

1. Why have a single prototype for each class? I would imagine having multiple prototypes would enhance coverage over the full diversity of class images.
2. What about outliers? When we take a mean over activations, we would effectively start to ignore outliers in our explanations.
3. Wouldn’t using the high frequency loss (no large difference in adjacent pixel values) smooth out the prototype image and, thus, cause the prototype to look more diffuse? I’m not sure if this is a good or a bad thing so I’m curious to know what the authors think.

---

> ### Author Response · Authors · 2023-11-21
>
> We thank the reviewer for their time and effort in this well-thought out review.
>
> We’ll start by addressing the questions:
>
> 1. Although multiple prototypes would cover all possible features that would contribute towards a particular class prediction, in this paper we were interested in exploring the potential of prototypes to cover the most relevant features of a class. We are very interested in adding robust diversity functionality in the prototype generation process and we have included this in our expanded future work section.
>
> 2. While it is true that taking the mean of activations would subdue outliers, this does not stop us from using the mean of activations to meaningfully compare our prototypes that aim to capture the most useful features in the average case with natural images. The notion of path similarity introduced by Geirhos et al 2023 [1] does fall into this trap of not appropriately addressing outliers.
>
> 3. The high frequency loss would lead to diffuse prototypes in isolation but it is an important objective to have in conjunction with all the other objectives in play during prototype generation. In our implementation, the high frequency loss helps smooth out adversarial jaggedness and artifacts rather than promote diffusion.
>
> Addressing the weaknesses:
>
> Although the prototypes require some subjective reasoning, they do enable development of hypotheses that can be effectively tested. Looking through a dataset of millions of examples is prohibitively impossible and even in the case of a reasonably sized dataset, hypotheses built purely on dataset analysis are boundless. Our prototypes help bound this process and guide us to actionable insights by directly querying the model.
>
> We do agree with the need for diversity and more natural-looking prototypes which we have added to our Future Work section.
>
> Our comprehensive experiments consist of comparing spearman similarity between prototype activations and natural image activations. Since we show high similarity we go ahead and show a case study of gathering insights from prototypes for two classes that seem interesting in our subjective analysis.
>
> Activation similarity is compared to counter the claims by Geirhos et al. 2023[1] , which demonstrated that visualisations using the established method have very unnatural activation paths. We include this measure against the mean natural image paths to demonstrate that our prototype generation method produces inputs that do have natural activations. We do not use this metric to generate our prototypes so it is possible that the prototypes can show outlier features in conjunction with the average features for a given class.
>
> [1] Robert Geirhos, Roland S. Zimmermann, Blair Bilodeau, Wieland Brendel, and Been Kim. Don’t trust your eyes: on the (un)reliability of feature visualizations, July 2023.

---

> > ### Comment · Reviewer_WSWR · 2023-11-23
> > **Response to Rebuttal**
> >
> > I thank the authors for their response.
> >
> > I do agree that this technique helps narrow down a space of possible explanations but it still does not provide a concrete explanation as has been stated in the paper and that is what I object to in my 'Weaknesses' section.
> >
> > The authors also haven't added details about their losses etc as I mentioned the review. I will, thus, maintain my score.

---

### Author Response · Authors · 2023-11-21

We thank the reviewers for their insightful thoughts and questions. We are uploading an updated version of our paper with three major changes:

1. We acknowledge our use of a diversity objective in Section 4
2. We show the limited results from the use of our diversity objective in Appendix C
3. We expand our future work section to add more work for the diversity objective and more robust prototype benchmarking including work highlighted by Casper et al. 2023 [1]

[1] Stephen Casper, Yuxiao Li, Jiawei Li, Tong Bu, Kevin Zhang, Kaivalya Hariharan, and Dylan Hadfield-Menell. Red teaming deep neural networks with feature synthesis tools, 2023

---

### Meta-Review · Area_Chair_d5Zj · 2023-12-14

**Metareview:**

The paper proposes to use feature visualization (akin to that by Nguyen et al. NeurIPS 2016) to visualize the class output neurons of image classifier.

The reviewers find that the paper has many writing issues e.g. reviewer `5oJb` and `fFHp` the manuscript fails to differentiate the proposed method from the typical feature visualization (Nguyen et al. NeurIPS 2016).
Furthermore, the work lacks a quantifiable metric that measures success of the generated images. Therefore, most important evaluation is through qualitative assessment.

Therefore, AC recommends `reject`.

**Justification For Why Not Higher Score:**

The work lacks quantitative evaluation of the utility of the generated images/results.
The manuscript fails to show the novelty of the proposed method vs. prior work (Nguyen et al. 2016).

**Justification For Why Not Lower Score:**

N/A

---

### Decision · Program_Chairs · 2024-01-16

Reject